# Pulmonary Function and Psychological Burden Three Months after COVID-19: Proposal of a Comprehensive Multidimensional Assessment Protocol

**DOI:** 10.3390/healthcare10040612

**Published:** 2022-03-25

**Authors:** Guido Vagheggini, Francesca Marzetti, Mario Miniati, Lorenzo Bernardeschi, Mario Miccoli, Giulia Boni Brivio, Simone Meini, Eugenia Panait, Elena Cini, Angelo Gemignani

**Affiliations:** 1Chronic Respiratory Failure Care Pathway, Department of Medical Specialties, Azienda Usl Toscana Nordovest, 56048 Volterra, Italy; 2Fondazione Volterra Ricerche Onlus, 56048 Volterra, Italy; eugeniapanait57@gmail.com (E.P.); angelo.gemignani@unipi.it (A.G.); 3Department of Surgical, Medical and Molecular Pathology, Critical and Care Medicine, University of Pisa, 56126 Pisa, Italy; francescamarzettimoro@gmail.com; 4Department of Clinical and Experimental Medicine, Section of Psychiatry, University of Pisa, 56126 Pisa, Italy; mario.miniati@med.unipi.it; 5Medical Special Unit for Continuity Care (USCA), Azienda Usl Toscana Nordovest, 56048 Volterra, Italy; lorenzo.bernardeschi1@gmail.com (L.B.); giuliabonibrivio@libero.it (G.B.B.); 6Department of Clinical and Experimental Medicine, University of Pisa, 56126 Pisa, Italy; mario.miccoli@med.unipi.it; 7Internal Medicine Unit, Felice Lotti Hospital, Azienda Usl Toscana Nordovest, 56025 Pontedera, Italy; simone.meini@uslnordovest.toscana.it; 8Pulmonary and Occupational Medicine Outpatient Service Volterra (PI), Azienda Usl Toscana Nordovest, 56048 Volterra, Italy; elena.cini@uslnordovest.toscana.it

**Keywords:** COVID-19, SARS-CoV-2, long COVID, respiratory function, psychological distress, lung diffusion

## Abstract

Persisting limitations in respiratory function and gas exchange, cognitive impairment, and mental health deterioration have been observed weeks and months after acute SARS-CoV-2 (COVID-19). The present study aims at assessing the impairment at three-months in patients who successfully recovered from acute COVID-19. We collected data from May to July 2020. Patients underwent a multidimensional extensive assessment including pulmonary function test, psychological tests, thoracic echo scan, and functional exercise capacity. A total of 21 patients (M:13; Age 57.05 ± 11.02) completed the global assessment. A considerable proportion of patients showed symptoms of post-traumatic stress disorder (28.6%), moderate depressive symptoms (9.5%), and clinical insomnia (9.5%); 14.3% of patients exhibited moderate anxiety. A total of eleven patients (52.4%) showed impaired respiratory gas exchange capacity (P-DLCO, DLCO ≤ 79% pred). Compared to patients with normal gas exchange, the P-DLCO subgroup perceived a significant worsening in quality of life (QoL) after COVID-19 (*p* = 0.024), higher fatigue (*p* = 0.005), and higher impact of lung disease (*p* = 0.013). In P-DLCO subgroup, higher echo score was positively associated with hospitalization length of stay (*p* = 0.047), depressive symptoms (*p* = 0.042), fatigue (*p* = 0.035), impairment in mental health (*p* = 0.035), and impact of lung disease in health status (*p* = 0.020). Pulmonary function and echo scan lung changes were associated to worsened QoL, fatigue, and psychological distress symptoms.

## 1. Introduction

As of 4 July 2021, more than 197 million confirmed cases of COVID-19 have been recorded worldwide, with a mortality rate of around 2.2% globally. Thousands of cases were hospitalized, and a high proportion required treatment in intensive care units (ICUs) [1]. Italy was one of the first countries to be hit heavily by COVID-19, starting in February 2020.

The acute clinical spectrum of COVID-19 has been well investigated with fever, cough, shortness of breath, and anosmia and ageusia as the most common symptoms.

A considerable proportion of patients may experience ongoing symptoms even months after the acute phase. Previous studies found persisting impairment in pulmonary function tests (lung volumes and respiratory gas exchange) in patients after COVID-19 [2], and changes were likely to be more pronounced in the subgroup of ICU survivors [3,4]. As previously observed in SARS-CoV and MERS-CoV infections, a high incidence of ICU-acquired weakness and cognitive impairment is expected, lasting several months in patients who survived to COVID-19 pneumonia [3]; however, post-acute symptomatology occurs not only in critically ill patients, but also in those with milder forms [5,6,7].

The long term sequelae of COVID-19, due to persistent symptomatology and prolonged organ dysfunctions, include chronic fatigue and dyspnea, breathlessness, mood and behaviors changes, insomnia, cognitive impairments, gastrointestinal and cardiological disorders, and musculoskeletal pain [8,9,10,11,12,13]. This phenomenon is commonly referred as Long COVID [14]. Moreover, mental health disorders, depressive or post-traumatic stress symptoms, anxiety, and sleep disturbances were reported [4,9,10,15,16,17,18,19,20], with an increased incident use of antidepressants and anxiolytic [4].

There is some emerging evidence of the duration, severity, and prevalence of Long COVID symptoms. The reported prevalence of any symptom among COVID-19 survivors at five weeks was 22.1% in a UK Office for National Statistics (ONS) survey, while the 12 week prevalence was 9.9% [21]. The recent study of Munblit et al. (2021) assessed the long term health and psychosocial consequences of COVID-19 in a cohort of 2649 patients six to eight months post-discharge and found that half of patients (47%) reported one or more persistent symptoms [9]. Another recent study involving outpatients with mild COVID-19 symptoms reported long term health consequences after 4 months in up to 27.8% of subjects [7].

As the SARS-CoV-2 virus gains entry into the cells via the angiotensin-converting enzyme 2 (ACE2) receptor, the virus can cause damage in a multitude of organs [22] that have the potential to undergo not only acute, but also chronic damage [23]. Some pathophysiological mechanisms of COVID-19 infection and immune response induced by the virus may be involved in determining multi-organ persistent sequelae. Neuropsychiatric sequelae as well as long term multiple organ dysfunctions (i.e., lungs, heart, and kidneys) could be induced by viral neuro-invasion [24] or by a virus-induced dysregulation of immune response [25] or by an endothelial injury [26]. The association between different organ dysfunctions needs to be further investigated in order to identify common pathophysiological pathways underlying a prolonged organ involvement.

The aim of this pilot study was to investigate, in a country (Italy) very early hit by COVID-19, the prevalence at three-months, in a cohort of patients who recovered from COVID-19 pneumonia, of persistent pulmonary dysfunctions and psychological burden, assessed through an original comprehensive multidimensional assessment protocol; moreover, risk factors and concomitant conditions associated with pulmonary and psychological abnormalities, and their relationship, were assessed.

## 2. Materials and Methods

### 2.1. Study Design and Participants

This single arm study was conducted between May and July 2020 on a cohort of patients, both hospitalized or not, who had COVID-19 pneumonia three months earlier, and were then referred for a follow-up visit to the Pulmonary Outpatient Clinic of Volterra Azienda USL Toscana Nordovest (Pisa, Italy).

All patients met World Health Organization (WHO) criteria for discontinuation of quarantine (no fever for 3 consecutive days, improvement in other symptoms, and 2 negative test results for severe acute respiratory syndrome coronavirus 2 (SARS-CoV-2) 24 h apart).

All patients who fully underwent the scheduled multidisciplinary evaluation protocol, according to our standard post-COVID multidisciplinary follow-up program, including pulmonary function tests (PFT), trans-thoracic echo graphic lung scan, and self-paced exercise test, were recruited in this study. All patients participated in a self-administered online survey to evaluate their Quality of Life (QoL), and health status and impact of COVID-19 on overall mental health. A written informed consent was completed by participants before enrollment, after receiving a detailed description of the study and having the opportunity to ask questions.

Participants were eligible if the following criteria were met: 1. a definite diagnosis of COVID-19 3 months earlier; 2. a reverse transcriptase-polymerase chain reaction for SARS-CoV-2 with a negative test result after infection; 3. the ability to understand and complete the informed consent.

Exclusion criteria were: age < 18 years, insufficient knowledge of Italian language, presence of an important neurological disorder, and severe medical concomitant conditions interfering with the study procedures. The Ethics Committee of the Azienda USL Toscana Nord-Ovest, Volterra (Italy) approved all the recruitment and assessment procedures.

### 2.2. Materials

#### 2.2.1. Multidisciplinary Evaluation

Biometrics, clinical characteristics, and general and COVID-19 clinical history were collected as well as results of pulmonary function tests, lungs echo scan, and sit-to-stand one-minute test (STS1’) as part of the routine multidisciplinary assessment.

Pulmonary function tests were performed according to European Respiratory Society Recommendations [27], using a spirometer Biomedin Baires (Biomedin^®^, Padova, Italy), according to recommendations for prevention of COVID-19 diffusion [28]. Reference values for single breath lung diffusion test (DLCO) for adult Caucasians were considered to define subgroups of patients [29].

Each patient underwent echographic evaluation of the lungs performed by a pulmonologist trained in trans-thoracic lung echography with a Mindray^®^, M7 model (Shenzhen Mindray Bio-Medical Electronics Co. Ltd., Shenzen, China) equipped with linear probe (10 MHz), with pleural pre-set, and a scanning depth of 6–7 cm (adjustable on chest dimension). According to a previously published protocol, eight areas per hemithorax were evaluated, recording a 4–5 s clip per each, with longitudinal position of the probe [30]. In each hemithorax target, six vertical lines defined 4 regions: parasternal line, anterior axillary line, posterior axillary line, scapularis line, and paravertebral line; each region was divided into a superior and an inferior area. An echo score ranging from 0 to 3 was assigned for each area; sub-pleural consolidations were recorded separately.

Each subject was asked to perform a one-minute sit-to-stand test (STS1′), and an exercise field test to evaluate exercise capacity and functional status, and to detect exercise-induced oxygen desaturation, dyspnea, and muscular fatigue. This test has been recently validated as an alternative test to the 6MWT in office practice and included in the evaluation of patients with interstitial lung disease [31]. Pulse oximetry, heart rate, respiratory rate, blood pressure, dyspnea, and fatigue Borg score were also recorded at baseline, immediately at the end of the exercise, and after 1- and 2-min during recovery [32].

#### 2.2.2. Psychological Impact

Socio-demographic and psychological data, information regarding COVID-19, and the perception of changes in health status were collected after discharge in an online structured electronic data collection system. All scales were self-administered and completed by all participants.

Participants were asked to evaluate on a visual analogue scale the following aspects: satisfaction about the support received from families and health care workers (from 0 “completely unsatisfied” to 10 “completely satisfied”), changes in QoL and health status after COVID-19 (from 0 “it is now worse than before” to 10 “it is now better than before”; 5 stated for “no significant changes”), and worries about the possibility of contracting COVID-19 again (from 0 “absolutely not worried” to 10 “absolutely worried”).

The following established scales were used to measure: depressive symptoms, anxiety, impairment in health status, resilience, insomnia, impact of events, and obsessive-compulsive signs and symptoms.

Zung’s Self-rating Anxiety Scale (SAS): The SAS is a 20-item self-report questionnaire to assess anxiety symptoms [33]. Participants are asked to evaluate how much the statements describe how they felt in the last week (0 = None, 4 = Most or all of the time). Total raw score ranges from 20 to 80, with higher scores indicating greater anxiety. The cut-off for an anxiety that is clinically significant is 40 [34].

King’s Brief Interstitial Lung Disease (K-BILD): The K-BILD is a 15-item self-report questionnaire to assess the health status of impaired patients with interstitial lung disease (ILD) in three domains: breathlessness and activities, psychological, and chest symptoms [35]. Participants are asked to evaluate how much the statements describe how they felt in the last two weeks with a 7-points Likert scalfor each item. K-BILD domains and total score were transformed to a range of 0–100, with higher scores indicating higher health status. K-BILD yields reliable scores in patients with ILD coefficient alphas of 0.90 [35].

Impact of Event Scale-Revised (IES-R): The IES-R is a self-administered questionnaire measuring the current subjective distress for a specific traumatic event [36]. This 22-item questionnaire is composed of three subscales: avoidance, intrusion, and hyperarousal. Participants are asked to report the degree of distress experienced for each item in the past 7 days. The 5 points on the scale are: 0 (not at all), 1 (a little bit), 2 (moderately), 3 (quite a bit), and 4 (extremely). An IES-R score ≥ 34 discriminates between individuals with and without clinically relevant posttraumatic stress disorder (PTSD) [37].

Insomnia Severity Index (ISI): The ISI is a 7-item self-administered questionnaire to assess insomnia symptoms and their severity. The total score ranges from 0 to 28 with a number of cut-offs: 0–7 = No clinically significant insomnia, 8–14 = Sub-threshold insomnia, 15–21 = Clinical insomnia (moderate severity), and 22–28 = Clinical insomnia (severe). [38].

Beck Depression Inventory-II (BDI-II): The BDI-II is a 21-item self-report questionnaire to measure depressive symptoms and their severity in subjects ≥ 13 years old [39]. Participants are asked to report on a 4-point scale the degree of severity of symptoms experienced in the last two weeks, in four domains: cognitive, affective, somatic, and vegetative. Items are rated from 0 “Not at all” to 3 “Extreme form of each symptom”. Global score ranges from 0 to 63 with the following suggested guidelines: minimal range = 0–13, mild depression = 14–19, moderate depression = 20–28, and severe depression = 29–63. [39].

Functional Assessment of Chronic Illness Therapy, Fatigue subscale (FACIT-F): The FACIT-F is a 13-items self-report questionnaire that assesses four dimensions of fatigue in patients with chronic illness: physical fatigue (e.g., I feel tired), functional fatigue (e.g., trouble finishing things), emotional fatigue (e.g., frustration), and social consequences of fatigue (e.g., limits social activity) [40]. Participants are asked to indicate if the statement applies to the patients’ situation in the past 7 days on a 5-point scale (from 0 “Not at all” to 4 “Very much”). The global score ranges from 0 to 52, with higher scores indicating better QoL.

12-Item Short-Form Health Survey (SF-12): The SF-12 is a self-reported outcome measure assessing the health impact on everyday life [41]. The SF-12 has two summary scores, mental health (MCS), and physical health (PCS) [42]. Total score ranges from 0 to 100 with lower scores indicating higher disability.

14-Item Resilience Scale (RS-14): The RS-14 is a 14-item questionnaire to assess the individual ability to withstand or adaptively recover from stressors [43]. Items are evaluated on a 7-point Likert scale ranging from 1 (strongly disagree) to 7 (strongly agree). Total score ranges from 14 to 98, with higher scores indicating greater resilience. [43].

### 2.3. Statistical Analysis

Data are presented as mean ± standard deviation (SD) or median with interquartile range (IQ), as appropriate. The comparisons between groups were performed using Mann–Whitney test. Spearman’s correlation was computed to evaluate the relationship between psychological distress variables, functional respiratory outcomes, echo scores, socio-demographic characteristics, and COVID-19 experience. *p* values < 0.05 were considered statistically significant. The analysis was conducted using IBM SPSS statistics, version 21.

## 3. Results

### 3.1. Demographic Characteristics and COVID-19 Experience

A total of 21 patients (M = 13; Age mean = 57.05 years, SD = 11.02) fulfilling the inclusion criteria were recruited for the study three months after COVID-19 recovery (days mean = 88.67, SD = 12.62).

A total of ten patients were hospitalized for an average of 10 days (SD: 14.42) and one patient (4.8%) was admitted to ICU. A total of eleven patients were treated at home.

A total of fifteen patients (71.4%) had pneumonia, 4 patients were oligo-symptomatic, and 2 patients (9.5%) had gastrointestinal symptoms. During the acute phase, ten patients required oxygen support (46.7%) and were hospitalized; three patients (14.3%) required high-flow nasal cannula, three patients (14.3%) received non-invasive ventilation, and one (4.8%) received invasive ventilation; only one hospitalized patient did not need oxygen therapy.

A total of eighteen patients never smoked, 3 patients were former smokers, and only one patient was a current smoker.

Demographic characteristics of the sample are summarized in Table 1.

Changes in health status or QoL vs. pre-COVID-19 were not perceived as consistent, even if some concern about the possibility to contract COVID-19 again seemed present (mean = 5.95, SD = 2.92). The majority of respondents were mainly satisfied with support received from their relatives (mean = 9.33, SD = 1.15) and from healthcare workers during hospitalization (mean = 7.71, SD = 2.83).

The characteristics of COVID-19 experience, and pharmacological and non-pharmacological therapies are summarized in Table 2 and Table 3, respectively.

### 3.2. Functional Respiratory Outcomes, Exercise Performances, and Lung Ultrasound

Table 4 summarizes the pulmonary function tests and lung ultrasound findings. Persistent gas diffusion impairment (DLCO < 80%pred) was found in half of participants (52.4%). Total Lung Capacity (TLC) restriction (TLC < 90%pred) was observed in 8/21 subjects (38.0%); airway obstruction in 5/21 (23.8%). Of 20 subjects, six showed an impaired performance in STS1’ test (n° of repetitions < 80%pred). A total of five subjects described a muscular fatigue (BORG_M) score > 5 points (23.8%) and 11 subjects (52.4%) reported a dyspnea (BORG_R) score > 5 points. A significant desaturation (>=3 units) occurred in 6/21 (28.6%) subjects during the STS1. No significant difference was found in previously hospitalized patients (*n* = 10) vs. patients treated at home (*n* = 11) for functional respiratory outcomes.

### 3.3. Outcomes of Mental Health in Total Cohort

All clinical variables were not normally distributed. A considerable proportion of patients had several symptoms of PTSD (IES-R ≥ 34, 28.6%), two out of 21 patients had moderate depressive symptoms (20 > BDI > 28, 9.5%) and clinical insomnia (ISI ≥ 22, 9.5%), and three patients had moderate anxiety, according to SAS (41 > SAS > 60, 14.3%). The median fatigue score was in line with that described in general population samples. The perceived physical health score was lower than expected in general population samples (43.6 vs. 55.7) but higher than expected in chronic obstructive pulmonary disease (COPD) patients (43.6 vs. 38.8). The same result was obtained for perceived mental health (49.6 vs. 53.9 in general population; 49.6 vs. 45.3 in COPD patients) [44]. Patients had high or very high resilience tendencies in 95.2% of the sample.

The median (IQ range) scores on the FACIT-F, BDI-II, SAS, ISI, IES-R, K-BILD, SF12 subscales, and RS-14 are shown in Table 5.

### 3.4. Respiratory Function and Outcomes of Psychological Distress: Differences between Subgroups

Following COVID-19, women perceived a worsening in their health status while men had no significant changes (2 vs. 5, *p* = 0.037).

There were two groups of patients defined according to lung diffusion capacity: pathological DLCO (P-DLCO) and normal DLCO (N-DLCO), if DLCO was <80% or equal or above 80% of predicted value, respectively. QoL of P-DLCO group worsened after COVID-19; no changes were found for patients with N-DLCO (3 vs. 5, *p* = 0.024). P-DLCO group also exhibited significantly lower scores in FACIT-F (38 vs. 45.5, *p* = 0.005) and K-BILD (62.2 vs. 87.2, *p* = 0.013) performances (Table 6).

#### Patients with Pathological DLCO

Correlation analyses in P-DLCO group showed that higher echo scores were significantly and positively associated with hospitalization (*p* = 0.047) and depressive symptoms (*p* = 0.042). Meanwhile, they were negatively related to VC (*p* = 0.041), SF12_MHS (*p* = 0.035), FACIT_F (*p* = 0.035), and K-BILD (*p* = 0.020). In N-DLCO group, a positive significant correlation was found only between echo scores and VC (*p* = 0.042). A positive and significant correlation was found for DLCO and VC (*p* = 0.021), FEV1 (*p* = 0.007), KCO (*p* = 0.019), anxiety (*p* = 0.033), and insomnia (*p* = 0.042) in P-DLCO group, while no significant correlation was found for DLCO in N-DLCO group. Correlations are summarized in Table 7.

## 4. Discussion

This study on mid-term sequelae of COVID-19 evaluated a broad range of physical and psychological effects assessed three months after recovery both in patients who were hospitalized or not.

As previously observed, we found Long COVID symptoms, such as persistent gas diffusion impairment and dyspnea, in more than 50% of participants, three months after discharge [9,13,15]. In our sample, these symptoms were observed also in mild COVID-19 home-treated patients, and we did not find any difference in functional respiratory outcomes between the two groups, as already reported [7,45].

Similarly to chronic post-SARS syndrome, Long COVID symptoms may include a range of psychiatric symptoms persisting months after the acute phase [46]. We observed symptoms of PTSD in about 30% of patients. Approximately 25% of patients had moderate anxiety and 10% of patients described clinical insomnia or clinically significant depression. These results are in line with previous studies on larger samples [10,16,17,18,19,47]. Moreover, we did not find significant differences in terms of mental health when comparing hospitalized patients vs. patients treated at home, nor comparing patients with normal vs. persistent impairment in lung diffusion capacity.

In the subgroup with impaired lung diffusing capacity, patients reported a reduction in QoL and an increasing fatigue significantly different from the other group. Moreover, the long-term abnormalities of echo score were associated with more frequent depressive symptoms, higher fatigue, and subjectively reduced psychological well-being. This result should be studied in larger samples in order to evaluate if these changes were determined by the lung function impairment or if a common pathophysiological pathway might underlie the two conditions.

One of the relevant mechanisms causing damage to the lungs and the respiratory tract in the acute SARS-CoV-2 infection is related to viral replication inside endothelial cells, thus resulting in endothelial damage and intense immune and inflammatory reaction. Chronic inflammation, fibroblasts activation, and prolonged activation of an hyperinflammatory and hypercoagulable state may contribute to the development of long term lung abnormalities and dyspnea. However, long term breathing difficulties after COVID-19 have been observed in subjects with no fibrotic lung abnormalities [48,49,50].

According to previous studies, we hypothesized that the impairment in diffusing lung capacity could suggest persistent abnormalities in gas exchanges related to long-lasting lung damage after COVID-19 pneumonia [51]. In the subgroup of patients with impaired diffusing lung capacity, we found restriction of lung volumes, but KCO was also reduced, indicating that the impairment in diffusion capacity was not directly related to the lung volume restriction, but other abnormalities may have contributed to it [52].

The relationships between VC and echo score in our sample showed concordance in P-DLCO subgroup, with higher scores in subjects with predicted lower VC. In N-DLCO subgroup this concordance was not found. The apparent opposite correlation in two subgroups should be further studied and verified in larger cohorts of subjects recovering from COVID-19 and with normal lung diffusion capacity. Lung ultrasound scan is affected by lack of specificity, and the echo graphic findings (B-lines and small lung consolidations) found in N-DLCO subgroup could be not linked to COVID-19 sequelae. A slightly altered lung ultrasound was frequently found in healthy COVID-19 negative subjects, and showed very poor specificity in poorly symptomatic COVID-19 subjects in a previous study [53]. Our finding supported the hypothesis that lung diffusion might show high sensibility and good specificity in the evaluation of post-COVID-19 lung functional impairment. However, even in subjects with normal gas diffusion (N-DLCO), long term lung fibrotic abnormalities and restriction of lung volumes may be observed, persisting for months [54,55].

### Limitations

Our study has some limitations that should be addressed. First, its cross-sectional design prevents us from drawing conclusions on the causal relationships of the functional respiratory outcomes observed after COVID-19, echo scores, exercise performances, and mental health. The design of this study allows only to describe the features of post-COVID-19 patients. Second, this single-center study with a small sample size limits the power of the study that should be intended as exploratory, in order to define the protocol to apply in a more extensive study. A subsequent study is currently ongoing with a larger sample size, aiming at confirming our results. Third, the lack of information on symptom history and psychological distress before acute COVID-19 should be considered, as it may increase vulnerability. Moreover, some patients may have developed additional comorbidities or complications, that may affect symptom prevalence and persistence, and the overall QoL.

## 5. Conclusions

Given the persistence of symptoms in patients that required hospitalization and ICU during the acute phase, but also in patients with mild COVID-19 symptoms, it is crucial to develop structured and interdisciplinary follow up and rehabilitation programs. Patients with sequelae of COVID-19 are increasing and they require an interdisciplinary approach, which integrates different disciplines for improving physical and mental health and QoL in the long term.

## Figures and Tables

**Table 1 healthcare-10-00612-t001:** Demographic characteristics.

*n* = 21		Mean (SD)—Range	*n* (%)
Age (years)		57.05 (11.02)–39–83	
Gender	F		8 (38.1%)
	M		13 (61.9%)
Days from infection		88.67 (12.62)–63–108	
BMI	Normal		12 (57.1%)
	Overweight		6 (28.6%)
	Obese		3 (14.3%)
Smoking status	No		17 (81.0%)
	Ex		3 (14.3%)
	Yes		1 (4.8%)
Education	Primary education level		6 (28.6%)
	Secondary education		10 (47.6%)
	Bachelor		3 (14.3%)
	Master, PhD, or equivalent		2 (9.5%)
Occupation	Self-employed		2 (9.5%)
	Employee		8 (38.1%)
	Retired		5 (23.8%)
	Housewife		2 (9.5%)
	Other		4 (19.0%)
Civil status	Married/with partner		16 (76.2%)
	Divorced		3 (14.3%)
	Widower		2 (9.5%)
Children	None		2 (9.5%)
	One		8 (38.1%)
	Two		11 (52.4%)
Sport	Never		8 (38.1%)
	Not yet but I used to perform sporting activities before COVID-19	4 (19.0%)	
	Yes, occasionally		1
	Yes, every week		8 (38.1%)

**Table 2 healthcare-10-00612-t002:** COVID-19 experience.

*n* = 21		Mean (SD)—Range	*n* (%)
COVID-19 spectrum	Pneumonia		15 (71.4%)
	Gastrointestinal		2 (9.5%)
	Pauci-symptomatic		4 (19.0%)
Treated at Home (*n*)			11 (52.3%)
Days of hospitalization		10.52 (14.42)–0–50	
Days in ICU	None		20 (95.2%)
	From 1 to 10 days		1 (4.8%)
Change in health status vs. Pre C19		3.86 (2.83)–0–9	
Change in QoL vs. Pre C19		4.76 (1.81)–2–9	
Concern about contracting C19 again		5.95 (2.92)–0–10	
Evaluation of family support		9.33 (1.15)–7–10	
Evaluation of health care workers		7.71 (2.83)–1–10	

**Table 3 healthcare-10-00612-t003:** Pharmacological and non-pharmacological therapies.

*n* = 21	*n* (%)
Lopinavir/Ritonavir	6 (28.6%)
Remdesivir	1 (4.8%)
Immunomodulants (Tocilizumab/antiJAK)	19 (90.5%)
Corticosteroids	4 (19%)
Antibiotic	14 (66.7%)
Heparin (prophylactic doses)	10 (47.6%)
Hydroxychloroquine	12 (57.1%)
Oxygen	10 (47.6%)
HFNC (high-flow nasal cannula)	3 (14.3%)
Non-invasive ventilation_	3 (14.3%)
Invasive ventilation_	1 (4.8%)

**Table 4 healthcare-10-00612-t004:** Functional respiratory outcomes, exercise performances, and lung ultrasound.

*n* = 21	Median (Interquartile Range-IQR)	*n* (%)
Tot. Charlson Score	4 (4–5)	
Cough		3 (13.6%)
TLC (% pred)	95 (82.5–108)	
FRC (% pred)	92 (72.5–104.5)	
VC (% pred)	107 (84–113.5)	
FEV1 (% pred)	101 (89–114)	
FEV1/VC (% pred)	104 (90.5–109)	
DLCO ((% pred)	79 (64–88)	
*Pathological (<80% pred)*		11 (52.4%)
*Normal*		10 (47.6%)
KCO (% pred)	81 (68.5–96)	
STS1_variation		
*Pathological (<80% pred)*		6 (28.6%)
*Normal*		14 (66.7%)
*Missing*		1 (4.8%)
BORG_R Score		
*Pathological (>5 points)*		11 (52.4%)
*Normal*		9 (42.9%)
*Missing*		1 (4.8%)
BORG_M Score		
*Pathological (>5 points)*		5 (23.8%)
*Normal*		15 (71.4%)
*Missing*		1 (4.8%)
ECHO SCORE	7 (2–11)	

**Table 5 healthcare-10-00612-t005:** Questionnaire scales and subscales scores and prevalence in total cohort.

*n* = 21	Median (IQR)	*n* (%)
FACIT-F	43 (38–47)	
BDI-II	4 (1.5–4)	
*Subthreshold symptoms*		19 (90.5%)
*Moderate depression*		2 (9.5%)
SAS	34 (30–38)	
*Subthreshold symptoms*		18 (85.7%)
*Moderate anxiety*		3 (14.3%)
ISI	4 (2–9.5)	
*No clinically significant insomnia*		15 (71.4%)
*Subthreshold insomnia*		4 (19%)
*Clinical insomnia (moderate severity)*		2 (9.5%)
IES-R	19 (10–36)	
*No clinically significant PTS*		15 (71.4%)
*Several symptoms of PTSD*		6 (28.6%)
K-BILD	76.6 (56.7–87.2)	
SF12_PCS	43.6 (35.9–54.3)	
SF_12_MCS	49.6 (42.2–52.8)	
RS14	81 (70.5–89.5)	
*Average*		1 (4.8%)
*High Resilience Tendencies*		10 (47.6%)
*Very High Resilience Tendencies*		10 (47.6%)

Abbreviations: FACIT-F = Functional Assessment of Chronic Illness Therapy, Fatigue subscale; SAS = Zung’s Self-rating Anxiety Scale; BDI-II = Beck Depression Inventory-II; ISI = Insomnia Severity Index; IES-R = Impact of Event Scale-Revised; K-BILD = King’s Brief Interstitial Lung Disease; SF12_PCS=12-Item Short-Form Health Survey, Physical Component Summary; SF12_MCS = 12-Item Short-Form Health Survey, Mental Health Component Summary; RS-14 = 14-item Resilience Scale.

**Table 6 healthcare-10-00612-t006:** Significant differences between subgroups.

*n* = 21	Median (IQR)		*p*-Value
	P-DLCO	N-DLCO	
	*n* = 11	*n* = 10	
TLC (% pred)	83 (77–95)	106.5 (94.75–118)	** 0.002 ** **
VC (% pred)	84 (82–107)	113.5 (104.75–124.75)	** 0.002 ** **
FEV1 (% pred)	91 (81–101)	107.5 (100.25–118.75)	** 0.006 ** **
KCO (% pred)	72 (65–86)	88.5 (77.5–103.75)	** 0.013 * **
Change in QoL vs. Pre C19	3 (3–5)	5 (4.75–7.25)	** 0.024 * **
FACIT_F	38 (31–47)	45.5 (42.25–48.25)	** 0.005 ** **
K-BILD	62.2 (50–78.8)	87.2 (75.5–90.5)	** 0.013 * **

Abbreviations: same abbreviations as Table 5. Bold data indicate correlations that are statistically significant. * *p* < 0.05; ** *p* < 0.01.

**Table 7 healthcare-10-00612-t007:** Significant Spearman correlation coefficients r_s_: associations between ECHO scores and DLCO, in P-DLCO vs. N-DLCO.

	P-DLCO	N-DLCO
*n* = 21	*n* = 11	*n* = 10
	r_s_	
ECHO Score		
Hospitalized	** 0.608 * **	−0.187
VC	** −0.622 * **	** 0.649 * **
BDI	** 0.621 * **	0.117
SF12_MHS	** −0.638 * **	−0.188
K-BILD	** −0.685 * **	0.459
FACIT-F	** −0.637 * **	0.071
DLCO		
VC	** 0.680 * **	−0.079
FEV1	** 0.755 ** **	−0.244
KCO	** 0.691 * **	0.296
SAS	** 0.642 * **	0.195
ISI	** 0.621 * **	−0.326

Abbreviations: same abbreviations as Table 5. Bold data indicate correlations that are statistically significant. * *p* < 0.05; ** *p* < 0.01.

## Data Availability

The raw data supporting the conclusions of this manuscript will be made available by the authors, without undue reservation, to any qualified researcher.

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
