# Peer review of "Pulmonary Function and Psychological Burden Three Months after COVID-19: Proposal of a Comprehensive Multidimensional Assessment Protocol"

_healthcare, 2022, doi:10.3390/healthcare10040612_

Round 1

Reviewer 1 Report

The manuscript describes an important global issue that clinicians should be aware of: the sequelae of Covid-19.

The introduction is clear and the methodology is well explained, but I have a doubt, when are all these scales filled in and where, at home, in the hospital, after discharge, during the stay? Were all scales self-administered or did a researcher, caregiver... supervise or assist the patient? Were all scales filled in at the same time? Please clarify this methodological aspect to improve the procedure.

"All clinical variables were not normally distributed" this is part of the results section.

In relation to the Impact of Event Scale-Revised (IES-R) or the Beck Depression Inventory-II (BDI-II), have you analysed the data of the patients who were hospitalised together with the data of the patients treated at home? Could this be a bias? Because these groups have had opposite experiences with the person who was hospitalised being worse. Although in the discussion section you mention these subgroups, you should take these differences into account for all analyses.

All participants completed all scales, some were missed. Confirm this (even if there are not).

As you mention, the small sample size is a limitation that prevents generalizations. More studies are needed.

Author Response

COMMENT

The manuscript describes an important global issue that clinicians should be aware of: the sequalae of Covid-19.

The introduction is clear and the methodology is well explained, but I have a doubt, when are all these scales filled in and where, at home, in the hospital, after discharge, during the stay? Were all scales self-administered or did a researcher, caregiver... supervise or assist the patient? Were all scales filled in at the same time? Please clarify this methodological aspect to improve the procedure.

ANSWER

Thanks for your comments. We modified the manuscript according to your suggestions. Please find at line 152-153 that all scales were self-administered after discharge and completed by all participants.

COMMENT

"All clinical variables were not normally distributed" this is part of the results section.

ANSWER

Following your suggestion, we moved the above mentioned  sentence in Result Section at line 257 (“All clinical variables were not normally distributed”).

COMMENT

In relation to the Impact of Event Scale-Revised (IES-R) or the Beck Depression Inventory-II (BDI-II), have you analyzed the data of the patients who were hospitalized together with the data of the patients treated at home? Could this be a bias? Because these groups have had opposite experiences with the person who was hospitalized being worse. Although in the discussion section you mention these subgroups, you should take these differences into account for all analyses. All participants completed all scales, some were missed. Confirm this (even if there are not).

As you mention, the small sample size is a limitation that prevents generalizations. More studies are needed.

ANSWER

Thanks for raising these points. As already specified (lines 152-153), all scales were self-administered after discharge and completed by all participants. Regarding IES-R and BDI-II, we didn’t’ find any significant difference between hospitalized patients and patients treated at home. We hypothesized that this was mainly due to the small sample size (N=10 vs N=11).

Reviewer 2 Report

Dear  collegues 

Thanks for the  paper .Just wondering about choosing the sample size would you like to comment on that ?

I advice the authors to discuss  the limitations  regarding the  few  number of cases. 

Congratulations.

Author Response

COMMENT

Dear Colleagues, Thanks for the paper. Just wondering about choosing the sample size would you like to comment on that?

I advice the authors to discuss limitations regarding the few number of cases. 

ANSWER

Thanks for raising this point. We are aware of the limitations in the generalizability of results, due to the small sample size. As we stated in the Discussion section (lines 363-366), this was a preliminary study in one center. Actually, we are going ahead with the recruitment of a larger sample, looking for soundly based results in more than one center.

Reviewer 3 Report

Following areas in his manuscript needs attention: 

  1. This is a single arm study and should be mentioned upfront in the methods
  2. The two groups instead to compare should be the ones requiring oxygen and the ones without. This groups should be analyzed for the variables defined
  3. Defining two groups post COVID based on DLCO brings questions to whether they had normal DLCO prior to COVID, because it is unclear if the DLCO reduction is truely due to COVID or few patients were heavy smoker or had ILD. Also authors should make effort to quantify their smoking such as Pack per years
  4. There are far too many abbreviations to keep track of in the manuscript which makes it difficult to read.
  5. In the manuscript there are two scores "Eco" and ECHO", are they different? please define them accurately
  6. Can patient baseline characteristics be clarified: such as heart failure, BMI, ILD or other lung disease, Anemia, because they tend to affect DLCO. Most literature suggests no change in DLCO in patient with Mild COVID, and so should be true in this study because majority were treated at home and/or required did not require oxygen
  7. Why did author include occupation, children, and education as a variable in table 1, because it may not be relevant to the study and if does have any relevance, authors should report one.
  8. Table 2 need formatting and the values are not aligned with the variables, such as "from 1 to 10 days" if off one line
  9. in section 2.1: authors excluded patients with "important neurological disorder and severe medical concomitant conditions interfering with the study procedures", please identify them and what were the exclusions
  10. The ECHO score was performed however it is unclear how many patients had lung diseases to begin with or prior to COVID, because that will affect the score. Authors should also report the level of training of the personnel performing ECHO
  11. Persistent diffusion impairment was found in half of patients, what were the severity of perfusion/DLCO impairment and were there any identifiable reasons (as suggested in point 6) for decrease in DLCO
  12. In the "limitation" section, authors state "some patients may have developed additional comorbidities or complications, that may affect symptom prevalence and persistence and the overall QoL", what were these identifiable complications? Since they were seen in the clinic, they should be easily identifiable.

Author Response

We thank the Reviewer for the time dedicated to the revision of this manuscript.

Please find below our point-by-point answers:

COMMENT

This is a single arm study and should be mentioned upfront in the methods.

ANSWER

Following your suggestion, we mentioned that it was a single arm study at line 97.

COMMENT

The two groups instead to compare should be the ones requiring oxygen and the ones without. These groups should be analyzed for the variables defined.

ANSWER

Thank you for raising this point: we didn’t’ find any significant difference between patients requiring oxygen and patients without. We hypothesized that this negative finding was due to the small sample sizes (N=10 vs N=11).

COMMENT

Defining two groups post COVID based on DLCO brings questions to whether they had normal DLCO prior to COVID, because it is unclear if the DLCO reduction is truly due to COVID or few patients were heavy smoker or had ILD. Also authors should make effort to quantify their smoking such as Pack per years ->

ANSWER

We thank the reviewer for the opportunity to clarify this point. All subjects were evaluated for smoking history and concomitant diseases. As shown in Table 1, 17/22 subjects were non-smokers (81%), two subjects (9.5%) were ex smokers and one was a current smoker (4.3%). The subjects reported no history of ILD.

COMMENT

There are far too many abbreviations to keep track of in the manuscript, which makes it difficult to read.

ANSWER

Following reviewer’s suggestion, we reduced as much as possible the number of abbreviations.

COMMENT

In the manuscript there are two scores "Eco" and ECHO", are they different? Please, define them accurately

ANSWER

Thank you for this point. The Eco and ECHO scores were the same. We revised the whole manuscript, and we used ECHO.

COMMENT

Can patient baseline characteristics be clarified: such as heart failure, BMI, ILD or other lung disease, Anemia, because they tend to affect DLCO. Most literature suggests no change in DLCO in patient with Mild COVID, and so should be true in this study because majority were treated at home and/or required did not require oxygen

ANSWER

Again, thank you for the opportunity to clarify this point. Patients were included in the assessment protocol after 63 to 108 days from infection. Giving that, our research team did not supervise the treatment of acute COVID. The treatment history was reported as resulting from patients’ chart records.

COMMENT

Why did author include occupation, children, and education as a variable in table 1, because it may not be relevant to the study and if does have any relevance, authors should report one.

ANSWER

The collection of the above-mentioned information aimed at providing an overall picture of socio-demographic variables and at excluding baseline differences between subgroups.

COMMENT

Table 2 need formatting and the values are not aligned with the variables, such as "from 1 to 10 days" if off one line.

ANSWER

Thank you, we modified Table 2 as suggested.

COMMENT

In section 2.1: authors excluded patients with "important neurological disorder and severe medical concomitant conditions interfering with the study procedures", please identify them and what were the exclusions

ANSWER

We decided to adopt, as exclusion criteria, those severe impairments of cognitive status that might prevent an active collaboration of the patient (e.g.: pulmonary function test, questionnaires completion), as well as the presence of commonly accepted contraindication to the exercise field test (Sit-to-stand1’) or to the forced vital capacity maneuvers.

COMMENT

The ECHO score was performed. However, it is unclear how many patients had lung diseases to begin with or prior to COVID, because that will affect the score. Authors should also report the level of training of the personnel performing ECHO

ANSWER

Thanks for this suggestion. We revised the manuscript following your indications (please, see lines 133-143) as follow:

“Each patient underwent echographic evaluation of the lungs performed by a pulmonologist trained in trans-thoracic lung echography with a Mindray®, M7 model (Shenzhen Mindray Bio-Medical Electronics Co. Ltd, Shenzen, P.R.China) equipped with linear probe (10 MHz), with pleural pre-set, and a scanning depth of 6-7 cm (adjustable on chest dimension). According to a previously published protocol, eight areas per hemithorax were evaluated, recording a 4-5 seconds clip per each, with longitudinal position of the probe [30]. In each hemithorax target six vertical lines defined 4 regions: parasternal line, anterior axillary line, posterior axillary line, scapularis line, para-vertebral line; each region was divided into a superior and an inferior area. An echo score ranging from 0 to 3 was assigned for each area; sub-pleural consolidations were recorded separately”

COMMENT

Persistent diffusion impairment was found in half of patients, what were the severity of perfusion/DLCO impairment and were there any identifiable reasons (as suggested in point 6) for decrease in DLCO

ANSWER

Lung diffusion capacity ranged from 64% to 88%, and transfer factor (KCO), from 68.5% to 96% of predicted values.

COMMENT

In the "limitation" section, authors stated that "some patients may have developed additional comorbidities or complications, that may affect symptom prevalence and persistence and the overall QoL", what were these identifiable complications? Since they were seen in the clinic, they should be easily identifiable.

ANSWER

Thank you for raising this point. We added this limitation because we followed our patients in the clinic for all functional tests, but the psychological questionnaires were completed at home in the following week. Some patients may have developed complications or reported symptoms at home that can affect symptoms’ prevalence and subjective QoL, and that might be not only related to Covid19 but also even with pre-existing conditions.
